

# Integrative bioinformatics analysis and experimental validation of key biomarkers driving the progression of cirrhotic portal hypertension

Meilin Li[1,*], Lilin Jiang[2,*], Yunrui Ru[3], Zhonghua Lu[4] and Peng Gu[5]

[1] Department of Gastroenterology, The Fifth People's Hospital of Wuxi (Affiliated Wuxi Fifth Hospital of Jiangnan University), Wuxi, China
[2] Department of Pathology, The Fifth People's Hospital of Wuxi (Affiliated Wuxi Fifth Hospital of Jiangnan University), Wuxi, China
[3] Experimental Center, The Fifth People's Hospital of Wuxi (Affiliated Wuxi Fifth Hospital of Jiangnan University), Wuxi, China
[4] Department of Hepatology, The Fifth People's Hospital of Wuxi (Affiliated Wuxi Fifth Hospital of Jiangnan University), Wuxi, China
[5] Department of Urology, Xishan People's Hospital of Wuxi City, Wuxi, China
* These authors contributed equally to this work.

Corresponding authors
Peng Gu, gdp_88@163.com
Zhonghua Lu,
Lu_z_h@jiangnan.edu.cn

## ABSTRACT

**Background:** Portal hypertension is a driving factor of cirrhosis complications, but the specific molecular mechanism of portal hypertension in cirrhosis remains unclear. The aim of this study was to identify hub genes for predicting persistent progression of portal hypertension in patients with liver cirrhosis.

**Methods:** Related microarray datasets were obtained from the Gene Expression Omnibus database. Weighted gene co-expression network analysis and differential expression genes analysis were used to identify the correlation sets of genes. In addition, protein-protein interaction networks and machine learning algorithms were conducted to screen center of candidate genes. To validate the diagnostic effect of hub genes, receiver operating characteristic curves were utilized in another dataset that is publicly accessible. Furthermore, the CIBERSORT algorithm was employed to investigate the immune infiltration levels of 22 immune cells and their connection to hub gene markers. Immunohistochemistry and reverse transcription quantitative polymerase chain reaction (RT-qPCR) were conducted to validate novel hub genes in clinical specimens.

**Results:** We obtained 671 differentially expressed genes and 11 module genes related to cirrhotic portal hypertension. Two candidate genes namely oncoprotein-induced transcript 3 protein (OIT3) and lysyl oxidase like protein 1 (LOXL1) were identified as biomarkers. RT-qPCR and immunohistochemistry (IHC) verified the expression of LOXL1 and OIT3 at mRNA and protein levels in liver tissue.

**Conclusions:** OIT3 and LOXL1 were identified as potential novel targets for the diagnosis and treatment of cirrhotic portal hypertension (CPH).

## INTRODUCTION

Portal hypertension (PH), defined as an increase in portal venous pressure, is a group of syndromes caused by a persistent increase in portal pressure for a variety of reasons and is a major consequence of cirrhosis (*de Franchis et al., 2022*). The pathophysiology of PH is complex, involving intrahepatic vascular resistance and extrahepatic hemodynamic changes. Current studies have shown that intrahepatic microcirculation disorder caused by liver sinusoidal endothelial cells (LSECs) dysregulation and hepatic stellate cells (HSCs) activation is the initial factor of portal hypertension in liver cirrhosis (*Iwakiri & Trebicka, 2021*). The changes of extrahepatic hemodynamics lead to the continuous worsening of PH. The hepatic venous pressure gradient (HVPG) remains the gold standard for assessing changes in portal vein pressure. HVPG exceeding 10 mmHg is clinically significant portal hypertension (CSPH), which represents an independent risk factor for the decompensation of liver cirrhosis, encompassing complications such as ascites, gastrointestinal variceal bleeding, hepatic encephalopathy, postoperative liver failure, and hepatocellular carcinoma. Unfortunately, as an invasive technique, HVPG has a limited clinical application. Consequently, the pursuit of non-invasive diagnostic biomarkers that are easily detectable and highly accurate for the diagnosis, and potentially for the treatment of cirrhotic portal hypertension (CPH), presents a formidable yet compelling challenge.

Recent advances in sequencing, mass spectrometry, bioinformatics, computational biology, and machine learning are revolutionizing biomedical research, making it possible to perform analyses of unprecedented scale and resolution using omics approaches (*Baker, 2023*; *Orlov, Baranova & Tatarinova, 2020*). Significant innovations in next-generation sequencing techniques and bioinformatics tools have impacted our appreciation and understanding of RNA, enabling utilization of comprehensive bioinformatics analysis, leveraging high-throughput sequencing, to identify new prognostic biomarkers associated with a range of diseases (*Li et al., 2022*). *Tian et al. (2023b)* identified TXNIP, CD44, and ENTPD1 as potential biomarkers for risk stratification in primary biliary cholangitis. *Kuang et al. (2024)* conducted transcriptome and metabolome sequencing on blood samples from patients with cirrhosis and healthy individuals, identifying potential common pathways in both the transcriptome and metabolome that could serve as candidates for future research. *Petrenko et al. (2024)* characterized the hepatic transcriptomes of two widely used liver fibrosis models, identifying transcripts that hold potential for development into prognostic biomarkers or consideration for therapeutic applications. Nevertheless, our understanding of the molecular mechanisms that drive PH remains inadequate.

By comparing gene expression profiles, researchers can identify specific genes or sets of genes that are differentially expressed in diseased tissue, which may serve as an indicator of disease progression or response to treatment. In this context, the application of bioinformatics not only enhances our understanding of the molecular basis of liver fibrosis, but also provides a basis for the development of targeted diagnostic and therapeutic strategies. Thus, our study sought to examine the differential gene expression between patients with cirrhotic portal hypertension (CPH) and healthy subjects, identify diagnostic

markers for CPH, ascertain the correlation between these biomarkers and immune cell levels, and investigate the underlying mechanisms to pave the way for future research.

## METHODS

### Data acquisition

We employed "Portal Hypertension" and "Liver Cirrhosis" as key search terms to identify Gene expression profiles in the Gene Expression Omnibus (GEO) database. Inclusion criteria for dataset selection: (1) Human liver tissue ≥6 samples per group; (2) availability of clinical metadata (excluded datasets lacking controls or with insufficient clinical metadata); (3) platform compatibility; (4) no batch effects detected after normalization.

Finally, the datasets GSE139602 (GPL13667) and GSE77627 (GPL14951) was accessed to download. GSE139602 consists of 39 samples: six control, eight compensated cirrhosis, and 12 decompensated cirrhosis. GSE77627 consists of 54 samples: 14 control and 22 cirrhosis, serving as a dataset for validation. We completed the data preparation using the R program (version 4.4.0; *R Core Team, 2024*). We removed probes corresponding to multiple compounds and converted probe IDs to gene symbols following the platform's annotation file. When multiple probes were associated with the same molecule, we retained only the one with the highest signal value.

### Differentially expressed genes analysis

Differentially expressed genes (DEGs) between the CPH (eight compensated cirrhosis, and 12 decompensated cirrhosis) samples and healthy control (HC) samples were determined using the R package limma. The thresholds were log2FC (fold change) > 1 and $p$ value < 0.05. The visualization of DEGs was conducted employing volcano plot and heatmap, utilizing the R packages ggplot2 and pheatmap.

### Weighted gene co-expression network analysis and key module genes selection

Weighted gene co-expression network analysis (WGCNA) represents a robust systems biology methodology, specifically engineered for the identification of co-expressed gene modules and for probing the connections between gene networks and significant phenotypic traits, as well as for elucidating the pivotal genes within these networks. In our study, we employed the WGCNA, executed using the R package WGCNA (*Yu et al., 2012*), to pinpoint modules exhibiting the strongest correlation with CPH. Initially, the top 25% of genes were chosen based on their variance. The R programming language's hclust function was utilized to conduct hierarchical clustering analysis, thereby identifying and excluding outlier samples. The pickSoftThreshold function ascertained an appropriate soft threshold β for the computation of intergenic adjacency. Subsequently, the adjacency matrix is utilized to transform the topological overlap matrix (TOM) as well as the associated dissimilarity measure (1−TOM). The hierarchical clustering tree was subsequently constructed to segment similar gene expression profiles into distinct modules. Finally, the expression profiles of the modules were aggregated by employing a

module-unique characteristic gene (ME), and the relationship between ME and clinical characteristics was quantified.

The Venn diagram was created using the online tool Draw Venn Diagram (https://bioinformatics.psb.ugent.be/webtools/Venn/) to overlap the genes in key modules and DEGs.

## Functional enrichment analysis

The Kyoto Encyclopedia of Genes and Genomes (KEGG) and Gene Ontology (GO) are established knowledge bases that serve as essential tools for the comprehensive analysis of gene functions and biological relationships. To conduct enrichment analysis on key genes, we used the R package "clusterProfiler" (*Yu et al., 2012*) to conduct GO and KEGG enrichment analysis, with the outcomes were visualized using R packages "ggplot2". A *p*-value less than 0.05 was deemed to be statistically significant.

## Protein–protein interaction network construction

Protein-protein interaction (PPI) network was constructed *via* STRING database (*Szklarczyk et al., 2019*), an online tool capable of identifying interactions involving genes or proteins. In practice, an interaction score of 0.400 was deemed the minimum threshold for medium confidence. To minimize potential interference and enhance network reliability, non-interacting genes were eliminated. Subsequently, the data pertaining to the interacting genes was transferred to Cytoscape (version 4.4.0) software for visualization. The pivotal module within the PPI network was pinpointed through the utilization of Cytoscape's plugin, Molecular Complex Detection (MCODE). This plugin is designed to cluster a specified network based on its topology, thereby identifying densely interconnected regions. Subsequently, the maximal clique centrality (MCC) algorithm of CytoHubba was used to explore the PPI network hub genes.

## Machine learning for screening hub genes

We employed three distinct machine learning algorithms—Least Absolute Shrinkage and Selection Operator (LASSO), Random Forest (RF), and support vector machine recursive feature elimination (SVM-RFE), to screen for key diagnostic biomarkers that predict the persistent progression of PH in patients with liver cirrhosis. LASSO regression is a classical computational learning method characterized by its ability to perform variable selection and regularization, which helps prevent overfitting and enhances the stability of linear models. RF, a non-linear classifier, adeptly identifies key predictors by uncovering the non-linear interaction relationships among variables, leveraging the decision trees. In the meantime, SVM-RFE is utilized to conduct iterative analyses on the entire set of genes, culminating in a gene ranking list ordered by their significance in terms of feature importance. In R environment, "glmnet" (*Friedman, Hastie & Tibshirani, 2010*), "randomForest" (*Tian, Wu & Yu, 2023a*) and "caret" (*Leiherer et al., 2024*) packages were utilized to develop the corresponding deep machine learning analyses. The candidate hub genes for diagnosis were selected based on the intersection of genes identified by LASSO, RF and SVM-RFE.

### Verification of hub genes expression and diagnostic efficacy

The expression level of hub genes was assessed employing the GSE139602 dataset. To assess the diagnostic precision of central genes, receiver operating characteristic (ROC) curves were constructed using the R package pROC (*Robin et al., 2011*) and displayed using the ggplot2 package. The aforementioned results were confirmed using the GSE77627 dataset. The diagnostic value was assessed by determining the area under the curve (AUC), where an AUC above 0.6 was regarded as an optimal diagnostic marker.

### Single-gene gene set enrichment analysis

Gene Set Enrichment Analysis (GSEA) for single-gene is a bioinformatics technique that leverages expression profiles to examine the functions and signaling pathways related to diagnostic biomarkers in particular biological processes and diseases (*Subramanian et al., 2005*). To achieve a more profound comprehension of the fundamental processes linked to the two biomarkers in CPH, the sample is divided into two different categories according to the median of OIT3 and LOXL1 expression level. The gene set designated as "c2.cp.v7.2.symbols.gmt" was retrieved from the Molecular Signatures Database (MSigDB) and chosen as the reference gene set. The threshold for statistical significance was established at a false discovery rate (FDR; q-value) of less than 0.05.

### Immune infiltration analysis

The CIBERSORT algorithm is a computational method that can convert a normalized gene expression matrix into a distribution of immune infiltration cells (*Steen et al., 2020*). Leveraging the LM22 as a benchmark for gene expression, the R package CIBERSORT was applied to analyze the immune cell expression patterns in both the CPH cohort and the HC group. Subsequently, Spearman's correlation was utilized to correlate the expression of immune cells and hub genes. Statistical significance was set at $p < 0.05$ in these correlations.

### Clinical sample collection

The liver tissue specimen of the CPH group was derived from patients with decompensated cirrhosis that underwent liver biopsy for diagnostic purposes to establish the etiology of the liver disease. The healthy liver tissue of the control group was derived from normal liver tissue surgically resected for benign liver tumors in our hospital. All participants in the study provided written informed consent, and the study protocol received approval from the Ethics Committee of Wuxi Fifth People's Hospital (approval number: 2023-0015-1).

### Quantitative RT-PCR analysis

The liver tissues from the healthy control and CPH group were subjected to thorough homogenization, followed by RNA extraction employing the TRIzol method. The RNA was then reverse transcribed into cDNA employing the Transcriptor First Strand cDNA Synthesis Kit (with gDNase, No. KR116; Tiangen Biotech, Beijing, China) according to the manufacturer's recommendations. Subsequently, we performed real-time PCR using FastReal qPCR PreMix (FP217; Tiangen, Beijing, China) under the following conditions:

**Table 1 Primer sequences information in the experiment.**

| Primer name | Primer sequence |
| --- | --- |
| GAPDH | Forward:GGAAGCTTGTCATCAATGGAAATC |
|  | Reverse:TGATGACCCTTTTGGCTCCC |
| LOXL1 | Forward:CAGCAGACTTCCTCCCCAAC |
|  | Reverse:CTGTGGTAATGCTGGTGGCA |
| OIT3 | Forward:GCCACTCTTGCCTTGGATCT |
|  | Reverse:CACAACACAGGGACTTGGCA |

pre-denaturation at 95 °C for 2 min, then pre-denaturation at 95 °C for 5 s, 60 °C 15 s, 40 cycles in total. The fold changes were calculated using the $2^{-\Delta\Delta Ct}$ method. The expression level is calculated and its standardization to endogenous GAPDH to determine. The primer sequences were shown in Table 1.

## Histopathology

Tissue samples were preserved in 4% paraformaldehyde and subsequently encased in paraffin. Liver samples were sliced into 4 μm thick sections and examined for tissue specimens *via* hematoxylin-eosin (H&E) staining. The immunohistochemistry (IHC) technique was utilized to identify the presence of LOXL1 and OIT3. IHC scoring utilized ImageJ software, analyzing images to determine IOD/Area values.

## Statistical analysis

Experimental data were presented as mean ± SD from a minimum of three separate trials. To compare differences between groups, we employed an unpaired t-test for normally distributed data and a non-parametric test for non-normally distributed data, respectively. Data analysis were conducted using GraphPad Prism version 8.0 and R software version 4.4.0 (*R Core Team, 2024*). Unless explicitly stated, the *p*-value less than 0.05 was considered statistically significant (*$p < 0.05$, **$p < 0.01$, ***$p < 0.001$, ****$p < 0.0001$).

## RESULT

### Identification of DEGs

The flowchart depicting the overall data screening strategy is presented in Fig. 1. First, we screened 671 DEGs (adjusted $p < 0.05$, |log2 FC| > 1) between the HC group and the CPH group in GSE139602 using the R package limma, where 241 genes were upregulated and 430 genes were downregulated. The DEGs were visualized as Volcano Plots and Heatmaps (Figs. 2A, 2B).

### Construction of co-expressed gene modules

In order to explore the potential correlation between disease and key genes, WGCNA was used to identify modules most significantly correlated with CPH. Module eigengenes clustering is utilized to visualize the outcomes of hierarchical clustering. In the diagram, "Height" signifies the degree of dissimilarity between clusters. When two clusters merge at

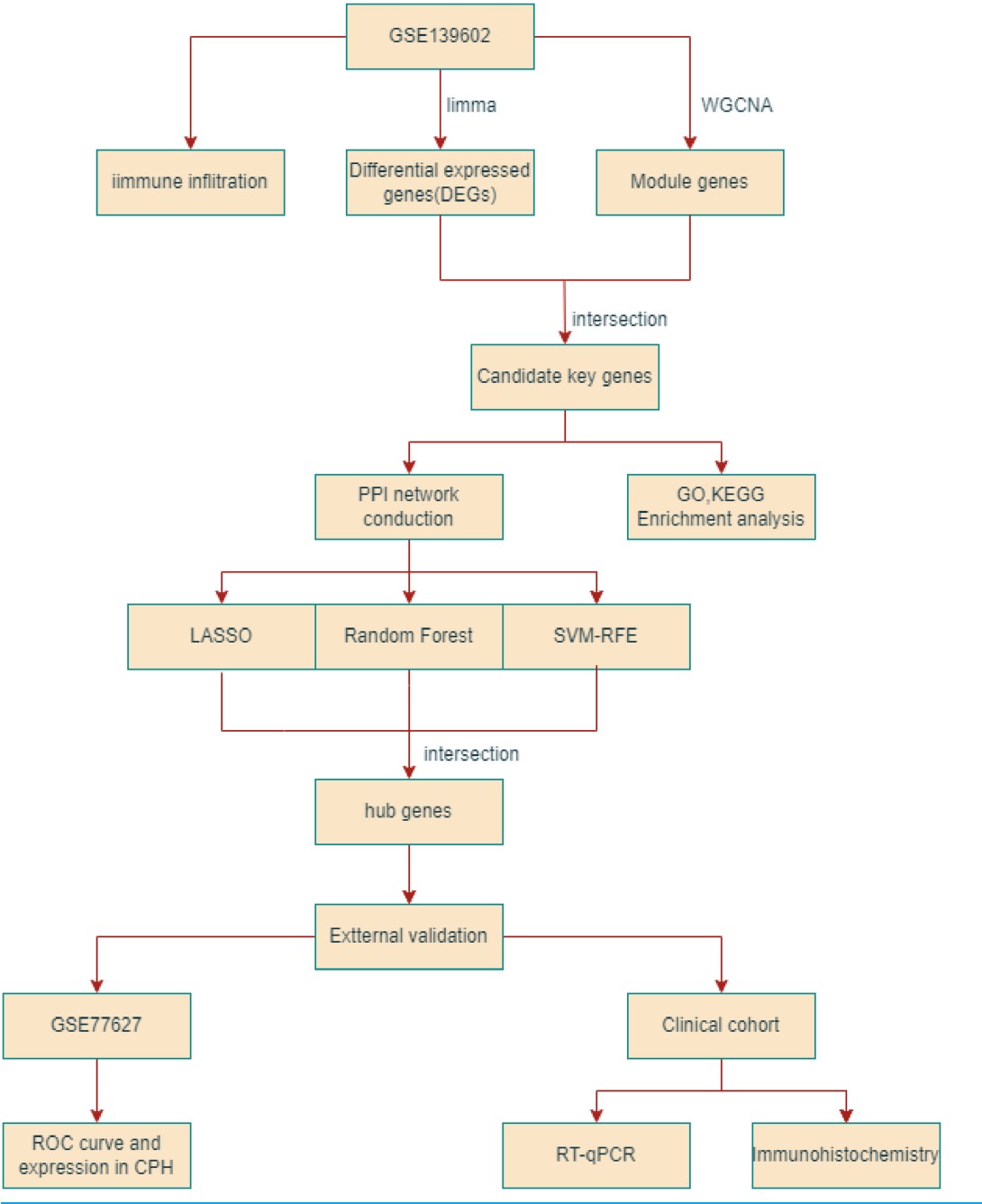

**Figure 1 The flowchart depicting the overall data screening strategy.**

a lower altitude, it signifies a greater degree of similarity between them; conversely, a higher merging point indicates a greater degree of dissimilarity (Fig. 3A). The soft-thresholding power (β) was established at 14 to ensure a scale-free R2 of 0.8, accommodating gene expression related to a scale-free network (Fig. 3B). The cluster dendrogram serves as a visual tool to illustrate the results of hierarchical clustering analysis. At the top of the dendrogram, a black line is present, where each branching point

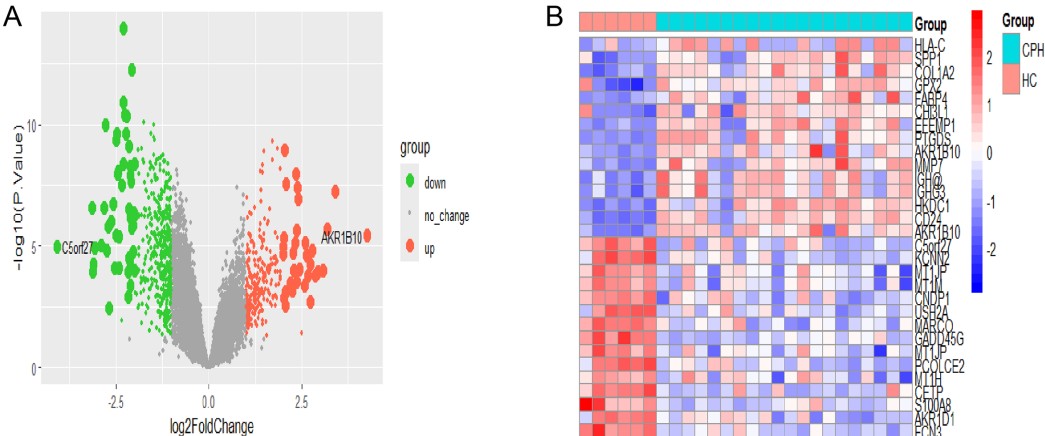

**Figure 2** **Identification of CPH-related DEGs.** (A) The volcano plot of DEGs in HC *vs* CPH group. (B) Heatmap displayed the top 30 genes that show significant differences.

signifies a division or combination in the clustering progression. Colored bands are used to indicate the distinct clusters identified by the Dynamic Tree Cut method, with each color corresponding to a specific cluster and the horizontal length of each band corresponding to the quantity of objects within that cluster (Fig. 3C). We identified a total of 11 modules in the module-feature relationship between the HC group and the CPH group, of which the green module (CC = 0.44, $p$ = 0.03) and the turquoise module (CC = 0.44, $p$ = 0.03) showed the most significant positive correlation (Fig. 3D). Upon integrating the genes from these two modules, we ultimately identified a total of 1,855 genes within the combined modules.

## Function enrichment analysis

To investigate the pathogenesis of CPH, we performed intersection of genes previously identified by DEGs and WGCNA. As shown in Fig. 3E, a total of 173 common genes were identified at the intersection. To delve into the molecular mechanisms of pathogenesis, shared genes were subjected to functional enrichment analysis employing both GO and KEGG methodologies. The GO analysis indicated that the key genes were predominantly enriched in several categories: (1) biological process (BP), encompassing extracellular matrix (ECM) organization, extracellular structure organization, external encapsulating structure organization, renal system development, and nephron development; (2) cellular component (CC), including collagen-containing ECM, basement membrane, collagen trimer, complex of collagen trimers and Golgi lumen; (3) molecular function (MF), including ECM structural constituent, growth factor binding, ECM binding, platelet-derived growth factor binding and ECM structural constituent conferring tensile strength (Fig. 4A). The KEGG pathway enrichment analysis revealed that shared genes were mainly concentrated in focal adhesion, PI3K-Akt signaling pathway, cytoskeleton in muscle cells, ECM-receptor interaction, protein digestion and absorption, human papillomavirus infection, complement and coagulation cascades and amoebiasis signaling pathways (Fig. 4B).

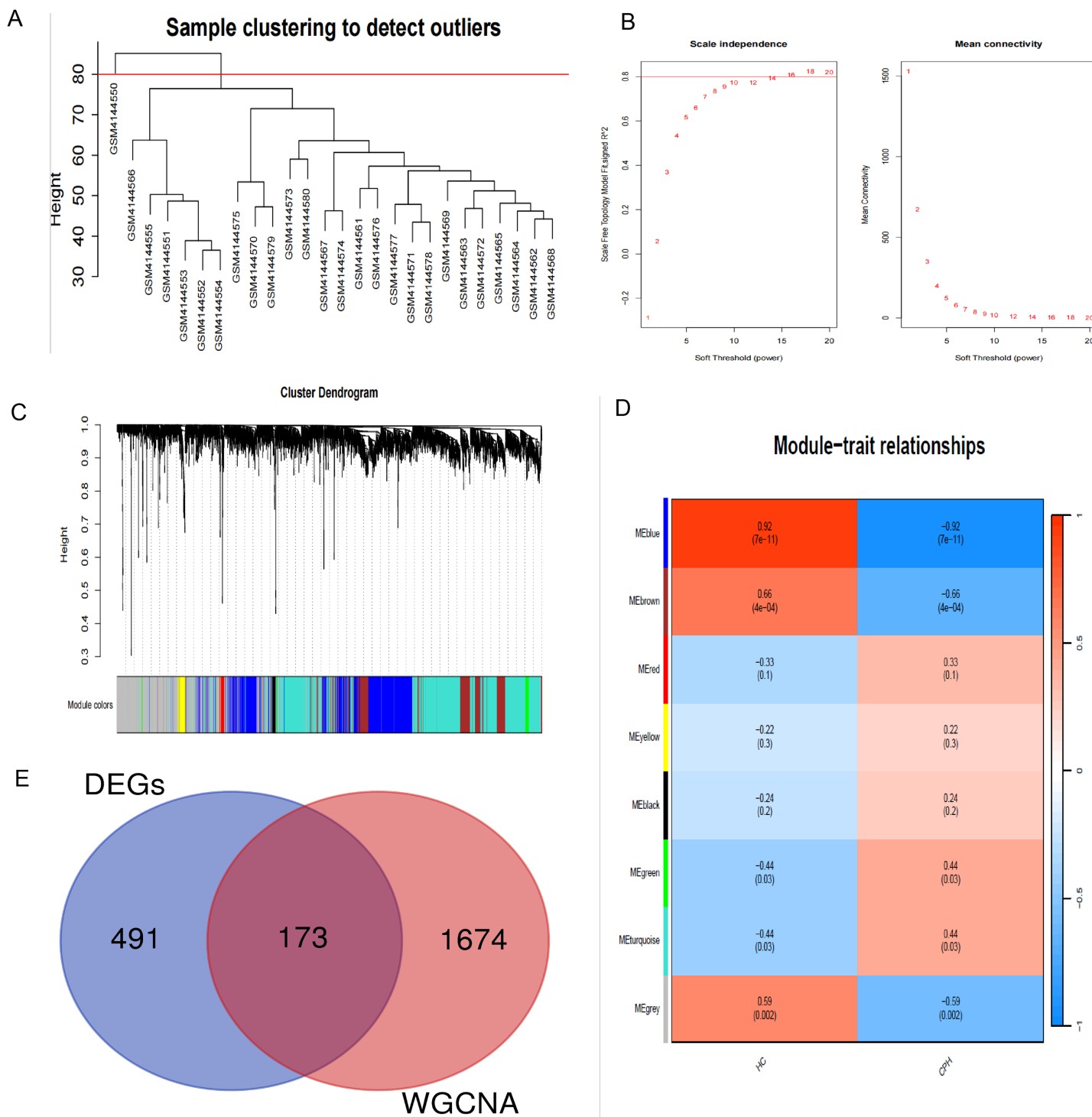

**Figure 3 Construction of WGCNA modules.** (A) Hierarchical clustering dendrogram of module eigengenes. (B) The "soft" threshold was chosen based on the combined analysis of scale independence and average connectivity. (C) The cluster dendrogram of co-expression network modules from WGCNA depending on a dissimilarity measure. (D) The CPH condition was characterized by 11 gene co-expression modules. Each cell within these modules displays the correlation coefficient and the corresponding *p*-value. (E) Shared genes identified by overlapping DEGs and WGCNA.

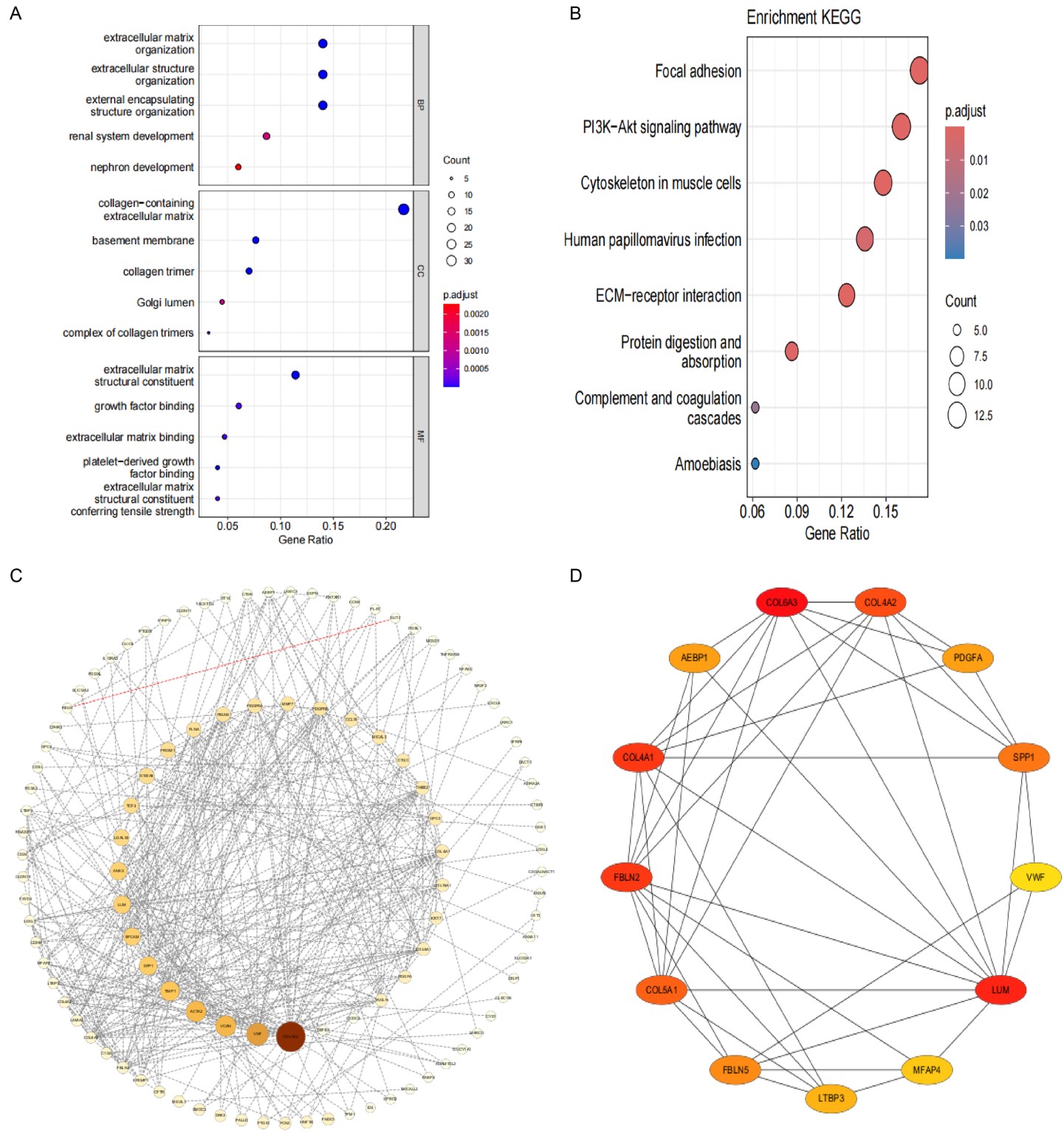

**Figure 4 Functional enrichment analysis and visual representation of the PPI networks.** (A) The enriched terms in GO analysis. (B) The KEGG enrichment analysis bubble plot displays the signaling pathways most closely related to 173 shared genes. (C) PPI network of interacted genes. (D) Gene clustering based on the MCODE algorithm.

## Analysis of the network of interactions between proteins

After eliminating 67 non-interactive genes, a PPI network was constructed using the remaining 106 node genes that exhibit mutual interactions (Fig. 4C). The MCODE algorithm was utilized to pinpoint the most significant cluster within the network, which encompasses 13 genes. Subsequently, employing the MCC algorithm, the interaction network of these 13 genes, consisting of 13 nodes and 41 edges, was derived using the CytoHubba plugin of the Cytoscape software (Fig. 4D).

## Identification of candidate hub genes *via* machine learning methods

In this research, we implemented three unique machine learning algorithms—namely, LASSO, RF and SVM-RFE—to pinpoint crucial hub genes linked to CPH. Regarding LASSO regression, we noticed that a mere seven genes were sufficient to attain the lowest binomial deviance on the curve (Figs. 5A, 5B). The RF lgorithm, which ranks genes according to their importance scores, identified 25 top candidate genes (Figs. 5C, 5D). In a similar vein, the SVM-RFE algorithm pinpointed peak accuracy when utilizing just two genes, guiding us to choose these as our candidates (Figs. 5E, 5F). In conclusion, we integrated the three aforementioned screening outcomes and identified the optimal gene signature, which includes two hub genes: OIT3 and LOXL1 (Figs. 5G).

## Expression and diagnostic efficacy identification of hub genes

The predictive and discriminatory powers of two principal genes, originating from the results mentioned above, were gauged by scrutinizing their expression patterns and executing receiver operating characteristic (ROC) curve analysis. In the preliminary phase, we scrutinized the expression levels of LOXL1 and OIT3 across the discovery cohorts (Figs. 6A). Upon examining the GSE193602 dataset, we found that the expression of LOXL1 was upregulated in CPH, while that of OIT3 was downregulated. This suggests that these two genes may have opposing roles at different stages of the disease. The diagnostic value of these two hub genes was further validated in the GSE193602 dataset using the ROC curve. In particular, both LOXL1 (AUC: 0.9273) and OIT3 (AUC: 0.9) exhibited substantial diagnostic significance for CPH, as depicted in Fig. 6C. Similar results were obtained in the GSE77627 validation dataset. Compared with the HC group, the expression pattern of OIT3 were descended in the CPH group, which consistent with that detected in discovery cohort. In addition, OIT3 (AUC: 0.8766) had good diagnostic efficacy. However, there was no significant difference in the expression pattern of LOXL1 between the two groups (Fig. 6B), the diagnostic efficacy of LOXL1 (AUC: 0.6169) was also unsatisfactory (Fig. 6D).

## Signaling pathways associated with hub genes

The signaling pathways pertinent to two key genes in CPH were identified through the application of GSEA. Figure 7A illustrates the set of upregulated genes associated with LOXL1. The top of the ranked list showcases a selection of enriched gene sets, encompassing pathways such as ECM-receptor interaction, focal adhesion, glycosaminoglycan biosynthesis−chondroitin sulfate/dermatan sulfate, intestinal immune

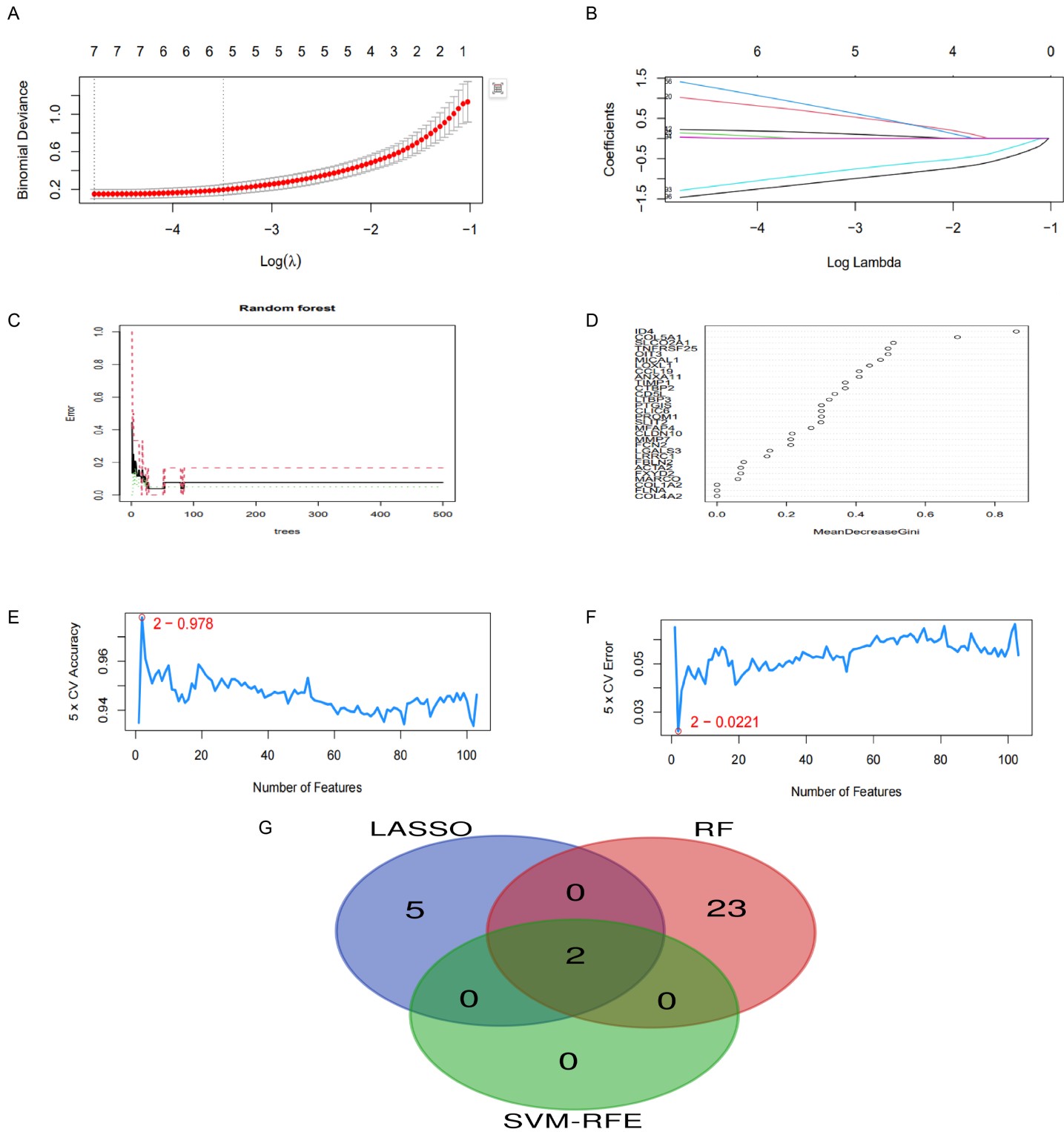

**Figure 5 Application of machine learning for screening hub genes.** (A, B) LASSO regression identified seven genes with minimal deviance. (C, D) RF algorithm selected 25 genes. (E, F) SVM-RFE algorithm highlighted two genes achieving peak diagnostic accuracy. (G) The intersection of three machine learning algorithms.

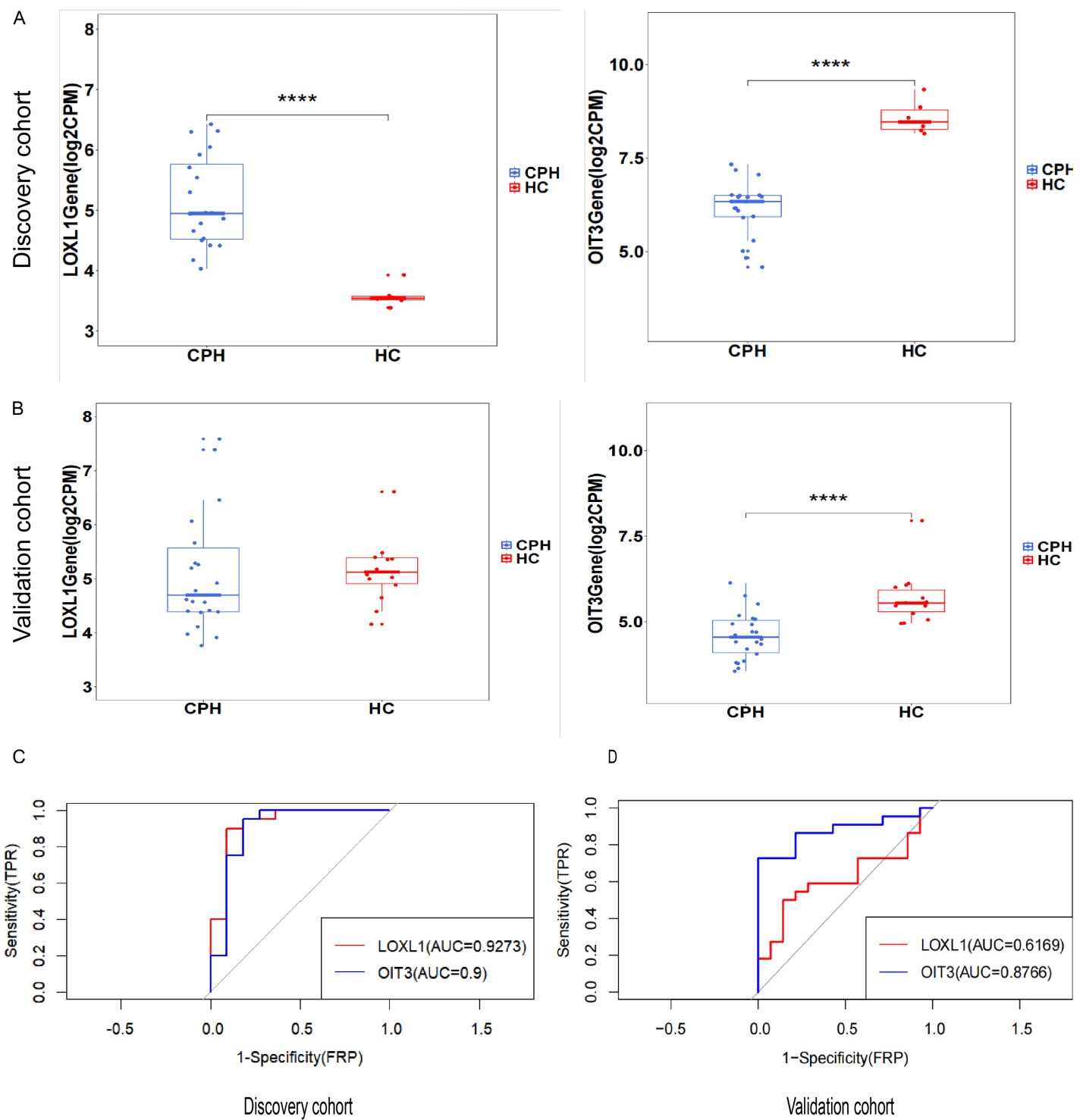

**Figure 6 Exploring the expression levels and predictive value of hub genes.** (A) Expression levels of two hub genes in the GSE139602: CPH group *vs.* HC group. (B) Expression levels of two hub genes in the GSE77627: CPH group *vs.* HC group. (C) ROC curve in the GSE139602. (D) ROC curve in the GSE77627. ****$p < 0.0001$.

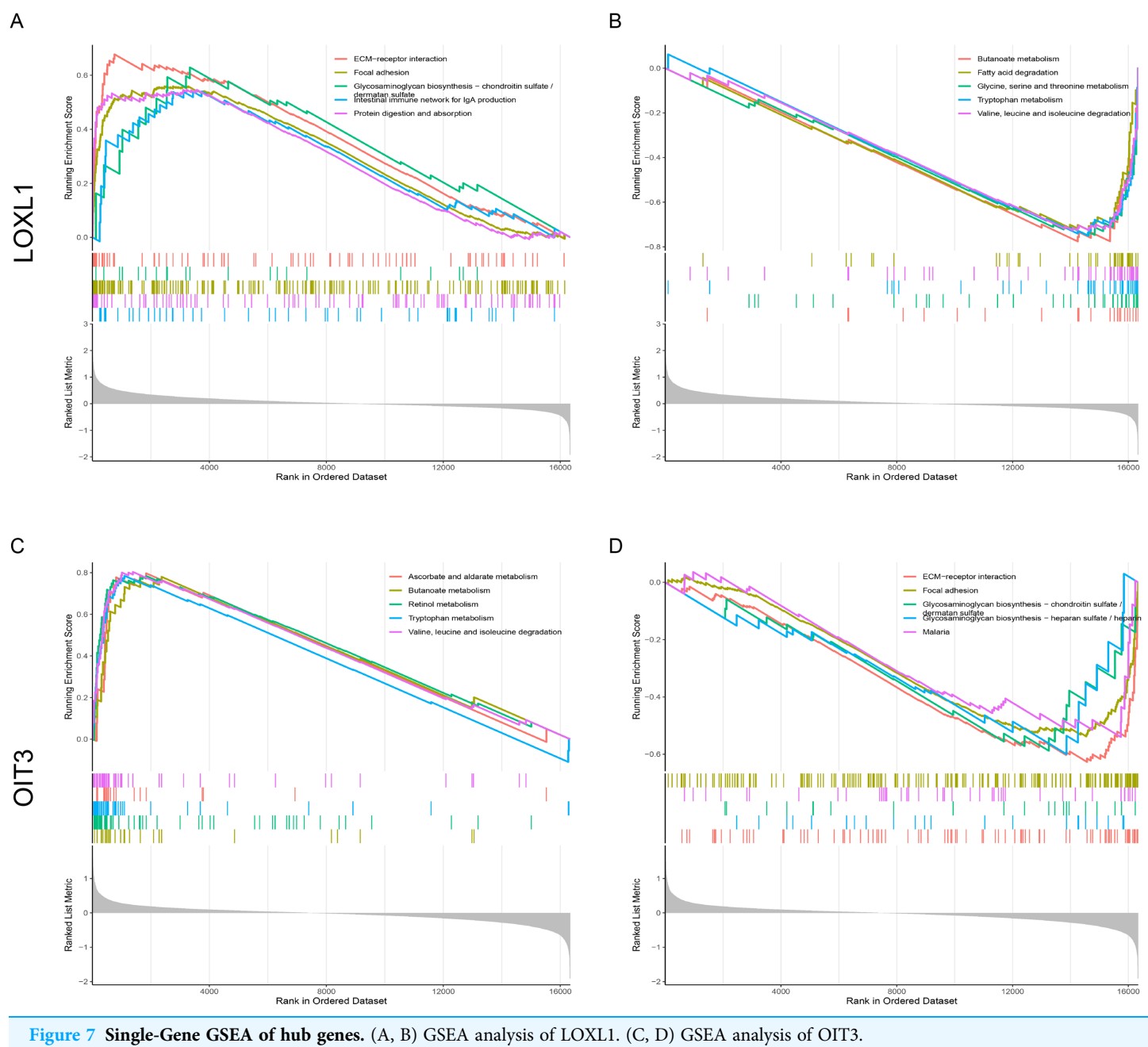

**Figure 7 Single-Gene GSEA of hub genes.** (A, B) GSEA analysis of LOXL1. (C, D) GSEA analysis of OIT3.

network for IgA production, and protein digestion and absorption. Figure 7B reveals gene sets corresponding to genes downregulated with LOXL1, highlighting butanoate metabolism, fatty acid degradation, glycine, serine and threonine metabolism, tryptophan metabolism, and valine, leucine and isoleucine degradation. Figure 7C illustrates gene sets related to upregulated genes linked to OIT3, encompassing ascorbate and aldarate metabolism, butanoate metabolism, retinol metabolism, tryptophan metabolism, and the degradation of valine, leucine, and isoleucine. Figure 7D presents gene sets associated with

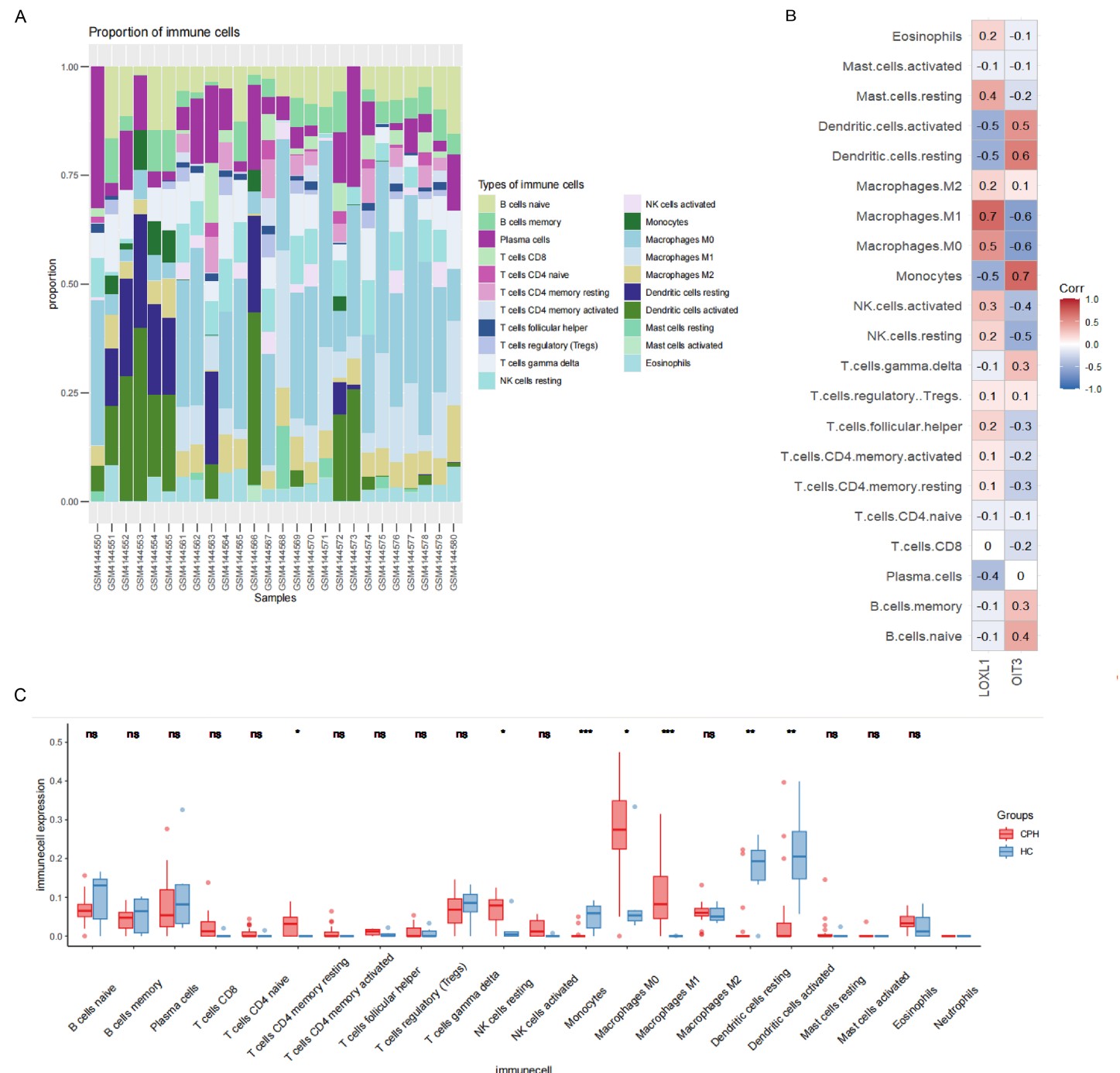

**Figure 8 Results of immune cell analysis.** (A) The stacked plot displaying the relative proportion of 21 immune cells in different samples. (B) The correlated heatmap between two hub genes and immune cells. (C) The box plot comparing the expression of immune cells between HC and CPH groups. *$p < 0.05$; **$p < 0.01$; ***$p < 0.001$.

downregulated genes linked to OIT3, including ECM-receptor interaction, focal adhesion, glycosaminoglycan biosynthesis–chondroitin sulfate/dermatan sulfate, glycosaminoglycan biosynthesis–heparan sulfate/heparin, and malaria. The findings suggest that the hub

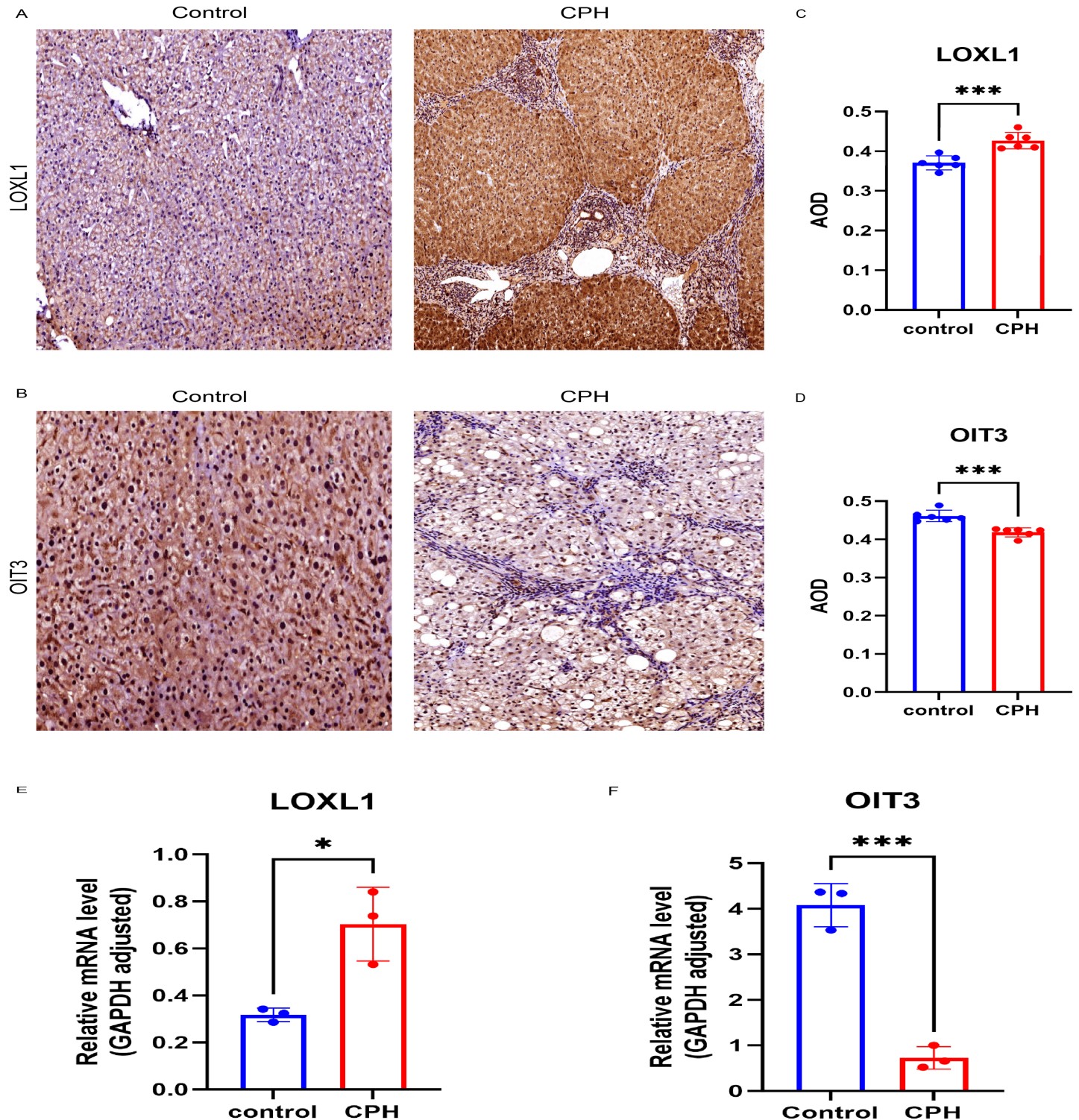

**Figure 9 Verification of the two hub genes.** (A) Immunohistochemical staining of LOXL1 in liver tissue. (B) Immunohistochemical staining of OIT3 in liver tissue. (C) The expression patterns of LOXL1. (D) The expression patterns of OIT3. (E) Relative mRNA levels of LOX1 in control and CPH group. (F) Relative mRNA levels of OIT3 in control and CPH group. Scale bar, 100 um. *$p < 0.05$; ***$p < 0.001$.

genes are strongly associated with the ECM-receptor interaction and focal adhesion pathways, potentially playing a crucial role in the pathogenesis and progression of liver cirrhosis with PH.

## Immune cell infiltration and its associations with diagnostic genes

In order to explore the role of immune cells in the progression of PH in cirrhosis, we used CIBERSORT algorithm to analyze the infiltration of immune cells. As illustrated in Fig. 8A, the relative proportions of 21 immune cell populations were represented through a stacked column graph. The CIBERSORT algorithm failed to detect the presence of neutrophils. Compared to the HC group, the CPH group exhibits a higher proportion of T cells CD4 memory resting, NK cells resting, macrophages M0, and macrophages M1, while having a lower proportion of monocytes, resting dendritic cells, and activated dendritic cells (Fig. 8C). The heatmap illustrating the correlation between genes and immune cells revealed that OIT3 exhibited a positive correlation with monocytes and dendritic cells and a negative correlation with macrophages and NK cells. Conversely, LOXL1 demonstrated a positive correlation with macrophages and mast cells, and a negative correlation with monocytes, dendritic cells, and plasma cells (Fig. 8B).

## Validation of hub genes expression

We examined the expression levels of LOXL1 and OIT3 in CPH and HC groups using RT-qPCR and immunohistochemical methods. In comparison to the HC group, the expression levels of LOXL1 were markedly increased in the liver tissues of individuals with CPH (Fig. 9A). In contrast, the expression of OIT3 was diminished (Fig. 9B), a result that was additionally supported by quantitative analysis (Figs. 9C, 9D). The results of the RT-qPCR analysis revealed significant differences in the expression of LOXL1 and OIT3 at the mRNA level between the control and CPH groups, aligning with previously predicted expression trends (Figs. 9E, 9F).

## DISCUSSION

Cirrhosis is currently the 11th leading cause of death worldwide, with approximately one million people dying each year due to complications from cirrhosis (Asrani et al., 2019). PH serves as the catalyst for complications in cirrhosis, precipitating life-threatening conditions and diminishing survival rates. Persistent liver damage, resulting from a range of causes, induces structural changes in the liver. This occurs through increased accumulation and turnover of the ECM—a process known as fibrosis—and alterations in cellular phenotypes. These changes are associated with the dysfunction of LSECs (Gracia-Sancho et al., 2021), HSCs, and inflammation from resident or infiltrating macrophages (Ortega-Ribera et al., 2023). These modifications cause a rise in intrahepatic resistance, which in turn boosts the pressure in the portal vein (resulting in portal hypertension), and this marks the first step towards the emergence of complications in chronic liver diseases. As a secondary consequence, PH triggers splanchnic and systemic arterial vasodilation, resulting in the emergence of a hyperdynamic circulatory syndrome. This, in turn, exacerbates and promotes clinically detrimental complications (Gunarathne et al., 2020).

HVPG remains the gold standard for assessing changes in portal vein pressure. Nevertheless, HVPG is invasive, difficult to conduct repeatedly, challenging to observe dynamically, and cannot be performed on patients with contraindications. Therefore, identifying a non-invasive examination method that can reflect the intrahepatic resistance of patients with cirrhosis and objectively predict the change of portal vein pressure is indispensable. Our goal is to enhance the comprehension of the physiological pathology and molecular mechanisms underlying CPH, thereby providing valuable insights for clinical diagnosis and treatment strategies. The environmental mechanobiology of PH has become an active research focus in recent year (*Felli et al., 2023*). *Gracia-Sancho, Marrone & Fernández-Iglesias (2019)* reveal CBX7, which was regulated by MiR-181a-5p was identified as a pressure-induced key transcription factor in LSEC and was correlated with HVPG. Meanwhile,the repression of two downstream targets by CBX7, specifically E-cadherin (ECAD) and serine protease inhibitor Kazal-type 1 (SPINK1), has been suggested as potential non-invasive biomarkers for PH (*Ortega-Ribera et al., 2023*). Consequently, this research aims to identify diagnostic biomarkers for CPH using high-throughput sequencing technology in conjunction with bioinformatics methods. It also seeks to explore the alterations that transpire as healthy individuals progress towards CPH.

In our study, we identified 671 DEGs through differential analysis of the gene expression matrix file from HC and CPH samples. Among these, 241 genes were upregulated, while 430 genes exhibited downregulated expression. The WGCNA technique was utilized to pinpoint genes exhibiting a high degree of co-expression and correlation with CPH, and 173 genes that coincide with the DEGs were chosen as possible candidates. The GO analysis indicated that 173 genes were primarily implicated in ECM organization, extracellular structure organization, basement membrane, growth factor binding and other biological functions. We revealed the signaling pathways that enriched these shared genes by KEGG analysis, including the interaction between focal adhesion and PI3K/Akt signaling. The findings indicate that abnormalities in genes associated with the ECM and basement membrane could play a role in the onset and progression of CPH. Additionally, the focal adhesion and PI3K/Akt signaling pathways may also be implicated. In the ongoing search for diagnostic biomarkers, the application of three machine learning algorithms was conducted: LASSO logistic regression, RF, and SVM-RFE. LASSO regression serves to decrease the likelihood of overfitting and chooses features that are not redundant. The RF algorithm captures non-linear interactions and ranks feature importance. SVM-RFE algorithm sequentially eliminates genes with minimal influence. Combining the above three machine learning algorithms can mitigate the bias brought about by the selection of a single model. Finally, two hub genes were obtained, namely, OIT3 and LOXL1. OIT3 and LOXL1 shows promise as a non-invasive biomarker for PH if its ROC performance is superior to existing markers. A retrospective single-center study from the emergency Department of the University Clinical Center in Serbia demonstrated the ROC curve analysis of AUC values of AST to platelet ratio index (APRI) and platelet-albumin-bilirubin (PALBI) is 0.603, and 0.606, respectively, for the prediction of CSPH (*Glisic et al., 2022*). In our study, OIT3's AUC is superior to APRI/PALBI, it could be a

better standalone test for PH. However, its clinical adoption depends on further validation, head-to-head comparisons, and integration into multi-parameter models. Research shows that there are systemic immune dysfunction in patients with cirrhosis and abnormal liver autoimmune reaction (*Albillos et al., 2022*; *Rodríguez-Negrete et al., 2024*). In this study, we further employed the CIBERSORT algorithm to assess the infiltration of 22 immune cell types within CPH samples. In comparison to the normal samples, the CPH samples demonstrated markedly increased levels of T cells CD4 memory resting, NK cells resting, macrophages M0, and macrophages M1. Monocytes and dendritic cells were positively correlated with the hub gene OIT3 and negatively correlated with the hub gene LOXL1, while macrophages were positively correlated with the hub gene LOXL1 and negatively correlated with the hub gene OIT3, indicating that immune cells play a role in the occurrence and progression of CPH.

Oncoprotein-induced transcript 3 protein (OIT3), also recognized as liver-specific zona pellucida domain-containing protein (LZP), is primarily found in hepatocytes and renal tubular cells. It is essential for the regulation of lipid metabolism, the transport of triacylglycerol, and the secretion of very low-density lipoprotein (VLDL) by the liver. Furthermore, it ensures the maintenance of urate balance by controlling the elimination and reabsorption of uric acid in the renal tubules (*Wu et al., 2021*; *Xu et al., 2004*; *Yan et al., 2012*). Moreover, new studies have indicated that OIT3 is a distinctive biomarker for macrophages with alternative activation and aids in the metastasis of hepatocellular carcinoma (*Yang et al., 2022*). It is currently acknowledged that elevated pressure within the portal venous system arises as a result of augmented resistance to the flow of portal blood. In the context of cirrhosis, this resistance is primarily dictated by microcirculatory dysfunction and structural irregularities, which are largely influenced by the compromised function of LSECs (*Gracia-Sancho, Marrone & Fernández-Iglesias, 2019*). OIT3, a potential marker gene that targets LSECs, may be essential in preserving liver equilibrium and shaping the pathological development of diverse hepatic disorders (*Li et al., 2023*). Here, our team reported that OIT3 was significantly downregulated in CPH samples compared with healthy samples. We assume OIT3 downregulation may disrupt LSECs mechanosensing, exacerbating portal resistance. However, its role in LSECs is still unclear, future desirable studies will help to clarify specific molecular mechanisms of OIT3 in maintaining mechanical homeostasis of LSECs and explore its potential as a novel molecular marker of CPH.

Lysyl oxidase (LOX) is a copper-dependent amine oxidase that belongs to a heterogeneous family of enzymes responsible for oxidizing primary amine substrates into reactive aldehydes. In mammals, the lysyl oxidase family consists of five members: LOX and four lysyl oxidase-like proteins (LOXL1-4). This enzyme family comprises copper amine oxidases, which possess a highly conserved catalytic domain, a lysine tyrosylquinone cofactor, and a copper-binding site that remains unchanged across members. They trigger the initial stage of the covalent cross-linking process involving ECM proteins such as collagens and elastin, which is crucial for the rigidity and mechanical integrity of the ECM (*Csiszar, 2001*; *Vallet & Ricard-Blum, 2019*). The reduction in LOXL1 expression led to a failure in the normal production of elastic fibers in

mice following childbirth, which in turn caused pelvic organ prolapse postnatally and a buildup of pro-elastin (*Liu et al., 2004*). *Ying, Chen & Yuan (2021)* determined that LOXL1 is crucial for the development of hypertrophic scars, and the suppression of LOXL1 can significantly impede the growth, movement, and accumulation of ECM in fibroblasts from hypertrophic scars by deactivating the Smad signaling pathway when exposed to TGF-β1. Recent research by *Hu et al. (2024)* revealed that the expression of LOXL1 is upregulated in rheumatoid arthritis and may reduce rheumatoid arthritis synovial inflammation by blocking the activation of the PI3K/AKT signaling pathway. Nevertheless, the clinical significance and biological functions of LOXL1 have not yet been documented in the context of CPH. In our study, our results showed that the expression of LOXL1 was upregulated in liver tissues of CPH. Combined with previous studies on LOXL1 related functions, it is reasonable to hypothesize that LOXL1 upregulation drives collagen cross-linking, perpetuating fibrosis and intrahepatic obstruction. The specific regulatory pathways of LOXL1 in the above pathogenesis will undoubtedly be the focus of future research.

As stated above, our in-depth bioinformatics analysis have pinpointed various signaling pathways and exposed a sequence of essential biological processes that are closely intertwined with the development of CPH. The dysregulation of these pathways and biological processes provides novel insights into the pathophysiological basis of CPH. However, this study has not yet performed thorough mechanistic validations of these results. Future research should aim to confirm the roles and significance of these genes in the progression of CPH by conducting functional experiments, including gene knock-out, overexpression studies, and immunohistochemical staining, both *in vivo* and *in vitro*. Furthermore, the existing research has solely focused on the analysis of microarray datasets at the transcriptional level, excluding the integration of genomic, proteomic, and metabolomic data. In the future, the comprehensive analysis of multi-omics data will provide deeper insights into the mechanisms behind disease development and progression.

## CONCLUSION

In conclusion, we believe that under various initiating factors including alcohol abuse, chronic viral hepatitis B or C infection, metabolic-associated fatty liver disease and so on, changes in core genes through complex regulatory networks lead to dysfunction of LSECs, activation of HSCs, and obstruction of intrahepatic microcirculation, thus driving the sustainable development of cirrhotic portal hypertension, leading to a series of complications, and ultimately reducing survival. We use bioinformatics methods to identify potential core genes through public databases, which promoted the occurrence and progression of cirrhotic portal hypertension. Through clinical validation of liver tissue, we found that hub genes OIT3 and LOX1 may be as potential therapeutic targets for CPH. However, our study was based on the public database, future studies should integrate multi-omics data and functional assays to validate mechanistic roles of OIT3/LOXL1 in CPH.

### Funding

This work was supported by Wuxi Municipal Health Commission Scientific Research Project (No. Q202306). The funders had no role in study design, data collection and analysis, decision to publish, or preparation of the manuscript.

### Grant Disclosures

The following grant information was disclosed by the authors:
 Wuxi Municipal Health Commission Scientific Research: Q202306.

### Competing Interests

The authors declare that they have no competing interests.

### Author Contributions

- Meilin Li conceived and designed the experiments, analyzed the data, prepared figures and/or tables, authored or reviewed drafts of the article, and approved the final draft.
- Lilin Jiang performed the experiments, prepared figures and/or tables, and approved the final draft.
- Yunrui Ru performed the experiments, analyzed the data, prepared figures and/or tables, and approved the final draft.
- Zhonghua Lu conceived and designed the experiments, authored or reviewed drafts of the article, and approved the final draft.
- Peng Gu conceived and designed the experiments, authored or reviewed drafts of the article, and approved the final draft.

### Human Ethics

The following information was supplied relating to ethical approvals (*i.e.*, approving body and any reference numbers):

The Ethics Committee of Wuxi Fifth People's Hospital approval to carry out the study within its facilities (approval number: 2023-0015-1).

### Data Availability

Raw data are available in the Supplemental Files.

Sequences available at NCBI GEO: GSE139602, GSE77627.

### Supplemental Information

Supplemental information for this article can be found online at http://dx.doi.org/10.7717/peerj.19360#supplemental-information.

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
