# Peer review of "Integrative bioinformatics analysis and experimental validation of key biomarkers driving the progression of cirrhotic portal hypertension"

_PeerJ, doi:10.7717/peerj.19360_

## Round 0.1 · original submission · Minor Revisions

Please address concerns of both reviewers and amend the manuscript accordingly.

**Language Note:** The review process has identified that the English language must be improved. PeerJ can provide language editing services - please contact us at [email protected] for pricing (be sure to provide your manuscript number and title). Alternatively, you should make your own arrangements to improve the language quality and provide details in your response letter. – PeerJ Staff

·

Basic reporting

--> The manuscript presents an integrative bioinformatics and experimental study on identifying novel biomarkers for cirrhotic portal hypertension (CPH).
--> The introduction provides a good background on portal hypertension but lacks a deeper discussion of recent advancements in bioinformatics approaches
--> Strengthen the literature review by discussing more recent studies applying bioinformatics
--> Ensure all figures are of high resolution.
--> Some figure legends lack sufficient detail (e.g., in Figure 5, clarify the significance of selected genes).
--> Ensure consistency in citation style. Some in-text references lack formatting.

Experimental design

--> Provide a clearer rationale for selecting OIT3 and LOXL1 as candidate biomarkers
--> The dataset selection process is unclear. How were GSE139602 and GSE77627 chosen? Were other datasets considered? Provide a justification for dataset selection, including inclusion/exclusion criteria
--> The choice of LASSO, Random Forest (RF), and SVM-RFE for biomarker selection is reasonable, but no justification is provided for selecting these methods over others. Briefly explain why these models were chosen and how they compare to alternative feature selection methods (e.g., XGBoost, deep learning approaches)

Validity of the findings

--> The explanation of the weighted gene co-expression network analysis (WGCNA) results is somewhat technical and lacks biological interpretation.
--> The ROC analysis needs a comparison with existing biomarkers to show if OIT3 and LOXL1 offer superior diagnostic performance. Provide more discussion on how OIT3 and LOXL1 relate to cirrhotic portal hypertension pathophysiology.
--> Compare the ROC curve results with existing liver disease biomarkers to demonstrate clinical relevance.

Additional comments

--> The study lacks a statement on code or script availability, which affects reproducibility. Consider sharing the code via a public repository (GitHub) and providing a link.
--> Improve clarity and language (proofreading required).
--> Justify dataset and machine learning model selection.
--> Provide more biological insights into OIT3 and LOXL1.
--> Compare ROC results with existing biomarkers.
--> Include a limitations section.

·

Basic reporting

1. Overall, the article is well-written, but it would improve with a thorough proofread by a native English speaker. Some sections need to be more concise and to the point.
2. The figures resolution can be increased.
3. Some relevant literature references have to be included.
4. The authors maintained an excellent article structures.

Experimental design

The research question is clear, pertinent, and significant. The way that research closes a recognized knowledge gap is explained.

Validity of the findings

Well-written conclusions that only support the findings and are connected to the original research topic.

Additional comments

1. Once the abbreviation is given for the words, the authors should use those abbreviations for the rest of the manuscript. The authors should thoroughly check this.
2. Check for the spellings throughout the manuscript.

---

## Round 0.2 · accepted · Accept

All issues indicated by the reviewers were addressed, and the revised manuscript is acceptable now.